# Paradox: Curcumin, a Natural Antioxidant, Suppresses Osteosarcoma Cells via Excessive Reactive Oxygen Species

**DOI:** 10.3390/ijms241511975

**Published:** 2023-07-26

**Authors:** Chunfeng Xu, Mingjie Wang, Behrouz Zandieh Doulabi, Yuanyuan Sun, Yuelian Liu

**Affiliations:** Department of Oral Cell Biology, Academic Centre for Dentistry Amsterdam (ACTA), Vrije Universiteit Amsterdam and University of Amsterdam, 1081 LA Amsterdam, The Netherlands; c.xu@acta.nl (C.X.); m.wang@acta.nl (M.W.); bzandiehdoulabi@acta.nl (B.Z.D.); y.sun@acta.nl (Y.S.)

**Keywords:** osteosarcoma, curcumin, apoptosis, ROS, NRF2

## Abstract

Osteosarcoma (OS) is an aggressive tumor with a rare incidence. Extended surgical resections are the prevalent treatment for OS, which may cause critical-size bone defects. These bone defects lead to dysfunction, weakening the post-surgical quality of patients’ life. Hence, an ideal therapeutic agent for OS should simultaneously possess anti-cancer and bone repair capacities. Curcumin (CUR) has been reported in OS therapy and bone regeneration. However, it is not clear how CUR suppresses OS development. Conventionally, CUR is considered a natural antioxidant in line with its capacity to promote the nuclear translocation of a nuclear transcription factor, nuclear factor erythroid 2 (NRF2). After nuclear translocation, NRF2 can activate the transcription of some antioxidases, thereby circumventing excess reactive oxygen species (ROS) that are deleterious to cells. Intriguingly, this research demonstrated that, in vitro, 10 and 20 μM CUR increased the intracellular ROS in MG-63 cells, damaged cells’ DNA, and finally caused apoptosis of MG-63 cells, although increased NRF2 protein level and the expression of NRF2-regulated antioxidase genes were identified in those two groups.

## 1. Introduction

Osteosarcoma (OS) is a primary bone malignancy (~20% of newly diagnosed bone tumors) with a low incidence (approximately 1–3 cases in one million), and it mostly strikes the adolescent and the senior citizen [1,2,3]. Although OS is rare, it is aggressive, and pulmonary and skeletal metastases are common in OS patients. Surgical resection combined with chemotherapy is the primary remedy for OS [4]. Extended resection is usually employed to prevent the local recurrence of OS, which may cause severe bone defects. Due to voluminous bone loss, it is highly universal for OS patients to suffer dysfunction, weakening the quality of life. Hence, bone rehabilitation cannot be ignored in comprehensive OS treatment. However, bone rehabilitation is still a massive challenge for OS patients after surgery because of the potential tumor relapse. Clinically, there is high demand for an agent with anti-cancer and pro-osteogenic properties in OS treatment.

Reactive oxygen species (ROS), the byproduct of normal oxygen metabolism in cells, including peroxides, superoxide, singlet oxygen, etc., are mainly generated in mitochondria [5,6]. They are involved in various physiological activities (stem cell renewal, cell differentiation, proliferation, etc.), while the over-physiological concentration of ROS can damage lipids, proteins, and DNA in cells [7], precipitating the apoptosis of cells [8]. On the other hand, a higher level of ROS than normal cells is one of the hallmarks of cancer [9], and the elevated ROS level is vital for the survival and proliferation of cancer cells [10,11]. Therefore, scavenging ROS and breaking the redox equilibrium in cancer cells might be an approach for cancer treatment. Nuclear factor erythroid 2-related factor 2 (NRF2), a nuclear transcription factor, plays a pivotal role in the development of several non-cancerous and cancerous diseases [12,13,14,15,16] through relieving intracellular ROS via the activation of target genes (*NADPH*, *HO-1*, *NQO1*, etc.) [17,18,19]. However, generally, NRF2 binds to Kelch-like ECH-associated protein 1(KEAP1), inhibiting its nuclear translocation and causing final ubiquitination [20], and the rapid ubiquitination leads to a low protein level of NRF2 in normal cells. Based on this, we assumed that an agonist of NRF2 that decreases ROS level in OS cells is likely to suppress OS development. On the other hand, it is also reported that NRF2 can induce oxidative stress via *KLF9* expression [21,22,23]. Therefore, the comprehensive role of NRF2 in ROS regulation should be further uncovered.

Curcumin (CUR), an extract from *Curcuma longa*, is generally recognized as a potent natural antioxidant [24,25], and it has been identified to be an agonist of NRF2 that can free NRF2 from KEAP1 and induce NRF2 nuclear translocation [26]. CUR can be used to treat various diseases, such as inflammation [27], cancer [28], a variety of pregnancy complications [29], and osteoporosis [30]. It has been validated that CUR can suppress OS [31,32]. In addition, it was reported that CUR might promote bone regeneration in OS treatment [33]. Regarding this, CUR may be a promising candidate for OS therapy with dual functions: OS suppression and pro-osteogeneration. CUR suppresses cancers through diverse signaling pathways [34]. Nevertheless, the mechanism of its anti-OS effect is still unclear.

This research was designed to elucidate the potential mechanism of the anti-OS effect of CUR. In line with the potent antioxidative effect of CUR and the vital role of ROS in cancer development, we previously hypothesized that CUR prevents NRF2 degradation, which leads to the decrease in ROS in MG-63 cells, causing the apoptosis of MG-63 cells. However, in this study, intracellular ROS in MG-63 cells were elevated with enhanced NRF2 nuclear translocation after CUR treatment, and the increased ROS induced the apoptosis of MG-63 cells. This result suggests that CUR may have different targets between OS cells and normal cells for ROS regulation, and CUR may have a dual character in the modulation of oxidative conditions.

## 2. Results

### 2.1. CUR Decreased the Viability of MG-63, Which Was Reversed by NAC

Figure 1 demonstrated the toxic effect of CUR on MG-63 cells and the rescue effect of NAC. After 24 h, 10 and 20 μM CUR significantly decreased the viability of MG-63 cells (*p* < 0.001), and NAC reversed this toxicity in these groups (*p* < 0.05). When MG-63 cells were treated for 48 h, 5 μM CUR was toxic to MG-63 cells (*p* < 0.05). Meanwhile, NAC rescued MG-63 cells in 5 and 10 μM CUR groups (*p* < 0.05 and *p* < 0.001, respectively). Moreover, with the extension of culture time, the suppression effect of CUR on MG-63 cells was also enhanced in three CUR-treated groups (*p* < 0.05, *p* < 0.01, and *p* < 0.05, respectively). This means that the toxicity of CUR to MG-63 cells is time- and dose-dependent. In addition, due to the rescue effect of NAC, a well-known antioxidant, it suggested that CUR-derived toxicity may be attributed to excessive ROS.

Up to 20 μM CUR demonstrated promising safety to DPSC (see Appendix A). This result shows that the toxicity of CUR is more specific to cancer cells.

### 2.2. CUR Elevated Intracellular ROS in MG-63 Cells

Results from the cell viability assessment implied that CUR increased ROS production in MG-63 cells, which was confirmed by ROS staining. ROS staining showed that 10 and 20 μM CUR dramatically induced ROS accumulation in MG-63 cells after 48 h (*p* < 0.01). On the other hand, NAC scavenged the excessive ROS in these two groups (Figure 2, *p* < 0.001).

### 2.3. CUR-Induced Oxidative DNA Damage

The oxidative damage of DNA was examined by the 8-OXOG staining. Enhanced 8-OXOG positive fluorescence was discovered in the 10 and 20 μM CUR groups after 48 h (Figure 3).

### 2.4. CUR Promoted the Apoptosis of MG-63 Cells

The results from the apoptosis assay also revealed that the toxic effect of CUR on MG-63 cells is time- and dose-dependent (Figure 4). Here, 20 μM CUR demonstrated outstanding toxicity to MG-63 cells when compared with other concentrations.

### 2.5. Effect of CUR on NRF2 Nuclear Translocation in MG-63 Cells

CUR-induced NRF2 nuclear translocation was detected using immunofluorescence staining (Figure 5). The result showed that 10 and 20 μM CUR dramatically stimulated NRF2 nuclear translocation in MG-63 cells after 48 h treatment.

### 2.6. CUR Modulated the Expression of Some NRF2-Related Genes

With the increase in NRF2, the expression of its downstream genes that regulate ROS production was also examined. The expression of *NQO1*, *SOD1*, and *HMOX1* showed an uptrend, although only *HMOX1* had a statistical difference (Figure 6, *p* < 0.001). However, the expression of *TXNRD2* was decreased (no significant difference). Moreover, 5, 10, and 20 μM CUR all activated the expression of *KLF9* (Figure 6, *p* < 0.001). However, this increase was inconsistent with ROS and NRF2 protein levels in these three groups.

## 3. Discussion

CUR has been reported to suppress OS development, but the mechanism remains unclear. This research was designed to explore this potential mechanism. Since CUR is a well-known natural antioxidant, and ROS plays an important role in cancer progression, it is reasonable to speculate CUR’s anti-OS ability relies on the decreased ROS in OS cells.

However, in this study, CUR-treated MG-63 cells had an increased ROS level, and the toxic effect of CUR was reversed by another antioxidant, NAC. This result implies that CUR suppresses OS cells via enhanced oxidative stress. This is contrary to our previous hypothesis. To determine whether CUR promoted or reduced the ROS accumulation, we detected the intracellular ROS level of MG-63 cells in this study. Results from the ROS assay demonstrated that CUR elevated ROS levels in MG-63 cells, and NAC mitigated this CUR-caused oxidative condition. The ROS assay results confirmed that CUR inhibits MG-63 cells through the enhanced ROS level, which is contrary to its role as an antioxidant. However, this result is consistent with some reported studies [35,36,37]. In these studies, CUR induced ROS production in different cancer cells. The inverse effect of CUR on ROS in cancer cells may be explained by the different CUR uptake between normal cells and cancer cells [38]. Since cancer cells contain more lipids on the cytomembrane, they can absorb more CUR, a natural polyphenolic compound, causing a higher concentration of CUR than in normal cells. Meanwhile, this partly explains why CUR used in cancer treatment is always modified with liposomes [39]. In addition, in this study, 5 μM CUR did not lead to ROS accumulation, which was also able to reflect that the effect of CUR on ROS is concentration-dependent: a high concentration could cause the overproduction of ROS. However, a low concentration may quench ROS.

NRF2 is a crucial protein for removing excess intracellular ROS. As an agonist of NRF2, CUR can promote nuclear translocation and avoid the ubiquitination of NRF2. Due to the CUR-caused ROS increment, we suspected that CUR did not induce NRF2 nuclear translocation in this research. Given this, we located NRF2 in CUR treated MG-63 cells using immunofluorescence. Nonetheless, 10 and 20 μM CUR stimulated NRF2 nuclear translocation in MG-63 cells. Then, we also analyzed the mRNA expression of some NRF2-target genes that are involved in ROS generation. The RT-qPCR results demonstrated the expression of some antioxidases genes was also enhanced. On the other hand, although *KLF9* expression was enhanced by CUR, the expression trend is inconsistent with that of ROS and the concentrations of CUR. These results suggest CUR may be involved in more complex and comprehensive cascades that modulate ROS production, not only the NRF2–ROS axis.

This study identified the oxidative DNA damage and apoptosis of MG-63 caused by CUR-induced ROS accumulation using 8-OXOG and annexin V/PI staining assay. The 8-OXOG was increased in MG-63 cells treated with 10 and 20 μM for 48 h. This means the abundant ROS caused by CUR has damaged the DNA of MG-63 cells. The annexin V/PI assay shows that the apoptosis of MG-63 cells is concentration-dependent and time-dependent. These outcomes confirmed that ROS-based cancer treatment is efficient. Based on this, photodynamic therapy (PDT) that inhibits cancer progression by drastically elevating ROS levels has been tested in various cancer models and shows promising results [40,41,42]. Meanwhile, CUR has been identified to be a photosensitizer [43,44] that can be used in the OS PDT treatment [31] to potent its anti-OS effect.

Although CUR can promote the apoptosis of MG-63 cells, its efficiency is unreasonable compared with some common chemotherapeutics. Walters et al. [45] treated seven OS lines with CUR for 72 h. The results showed the IC50 of CUR ranged from 14.4 to 24.6 μM. Nevertheless, the IC50 of cisplatin was only 1.24 and 3.65 μM for MG-63 and SaoS-2 cells, respectively, after 48 h treatment [46]. The high IC50 of CUR may be a result of its unstable bioactivity and low bioavailability. Despite that, CUR can suppress OS cells and promote bone regeneration, making it a promising candidate for the treatment of OS. Moreover, CUR in cancer treatment can reverse chemotherapy resistance and reduce the cytotoxicity of chemotherapeutics to normal cells [47,48,49]. Hence, a strategy that combines CUR with other treatments in OS treatment has been proposed and achieves prospective results [21]. Another advantage of CUR in OS treatment is its cancerphilia, which has been testified to in this research. With the same concentration, CUR is more specific to cancer cells. This means CUR is safer than chemotherapeutics and radiotherapy.

In this study, CUR promoted the apoptosis of MG-63 cells by amplifying intracellular ROS, rather than eliminating ROS in vitro, and the CUR–NRF2–antioxidases axis did not reverse the oxidative condition in MG-63 cells. As mitochondria are the main organ where ROS are generated, we will further expand our research on the effect of CUR on alterations in mitochondria’s functions and structures.

## 4. Materials and Methods

### 4.1. Cell Lines and Culture

Dental pulp stem cells (DPSCs) were from our lab. The MG-63 cell line was purchased from the American Type Culture Collection (ATCC, Manassas, VA, USA). After thawing, these cells were maintained in Minimal Essential Medium α (Gibco, Billings, MT, USA) supplemented with 10% fetal bovine serum (Gibco, Billings, MT, USA), 100 μg/mL streptomycin, and 100 μg/mL penicillin (Sigma-Aldrich, St. Louis, MO, USA) in a 5% CO_2_ incubator at 37 °C. A total of 100 mM CUR (Sigma-Aldrich, St. Louis, MO, USA) in dimethyl sulfoxide (DMSO, Sigma-Aldrich, St. Louis, MO, USA) was prepared as a store solution and stored in the dark in a −20 °C fridge. Finally, 5, 10, and 20 μM CUR and 5 mM N-Acetyl-L-cysteine (NAC, Sigma Aldrich, St. Louis, MO, USA) with relative concentrations of CUR were administrated to treat DPSCs and MG-63 cells.

### 4.2. Cell Viability Assay

PrestoBlue™ HS cell viability reagent (Thermo Fisher, Waltham, MA, USA) was used to detect the viability of CUR-treated DPSCs and MG-63 cells. According to the manufacturer’s instructions, cells were seeded in a 96-well culture plate at a density of 5000 cells/well (3 duplicates in each group). After attachment, these cells were treated with 5, 10, and 20 μM CUR with or without 5 mM NAC for 24 and 48 h. Afterwards, the medium was discarded, and 100 mL mixture of PrestoBlue™ HS and the culture medium (1:9) was added in each well. After incubation for 2 h at 37 °C, the supernatant was transferred to a new 96-well plate, and the fluorescence of each well was measured using a SpectraMax fluorescence multi-model plate reader (Molecular Devices, San Jose, CA, USA).

### 4.3. Quantitation of Introcellular ROS Level

To detect the intracellular ROS level, MG-63 cells were seeded in a dark-wall 96-well plate (1.0 × 10^4^ cells/well) at 37 °C with 5% CO_2_. After 12 h, cells were treated with corresponding reagents for 24 and 48 h. The medium was replaced with 100 μL of ROS assay working solution (Abcam, Cambridge, UK). After incubation at 37 °C for 60 min, cells were rinsed three times with PBS. A fluorescence microscope (Leica, Wetzlar, Germany) collected the images. Fluorescence was quantified with the Fiji software (version 2.14, NIH, Bethesda, MD, USA).

### 4.4. Immunofluorescence Staining of 8-OXOG

To detect ROS-caused oxidative DNA damage, MG-63 cells were cultured in a dark-wall 96-well plate (1.0 × 10^4^ cells/well). After attachment, cells were treated with 5, 10, and 20 μM CUR for 48 h. Immunofluorescence staining was used to detect 8-oxoguanine (8-OXOG), a marker of oxidative damage of DNA [50,51]. Briefly, after being rinsed with cold PBS, cells were fixed by 4% formaldehyde for 10 min and permeabilized in PBS containing 0.05% Triton X-100 for 5 min at room temperature. Cells were blocked by BlockAid™ Blocking Solution (Invitrogen, Waltham, MA, USA) for 1 h at room temperature. Thereafter, cells were incubated with the primary antibody diluted in blocking buffer (1:50) at 4 °C overnight. On the second day, cells were washed with cold PBS for 3 × 10 min and incubated for 1 h at room temperature in the dark with Alexa Fluor^®^ 594 conjugated secondary antibody (Abcam, Cambridge, UK). Then, 1 μg/mL Hoechst 33342 was used to stain the nuclei of MG-63 cells. Finally, cells were rinsed with cold PBS three times. Images were acquired using a fluorescence microscope. Fluorescence intensity was assessed with the Fiji software (version 2.14, NIH, Bethesda, MD, USA).

### 4.5. Quantitation of Cell Apoptosis

The apoptosis of CUR-treated MG-63 cells was detected by an annexin V and propidium iodide (PI) assay kit (BD Bioscience, Franklin Lakes, NJ, USA). MG-63 cells were treated with CUR for 24 and 48 h. Then, cells were trypsinized and washed twice with cold staining buffer to remove the remaining trypsin. Afterwards, 1.0 × 10^7^ cells were resuspended in 100 μL annexin V binding buffer followed by the addition of annexin V and PI (5 μL, respectively). These cells were generally vortexed and cultured for 15 min at room temperature in the dark. Before detection using a flow cytometry (BD FACSCalibur, San Jose, CA, USA), 400 μL of binding buffer was supplied in each group.

### 4.6. Immunofluorescence Staining of NRF2

To examine the efficiency of CUR on NRF2 nuclear translocation, MG-63 cells were treated with CUR in a dark-wall 96-well plate (1.0 × 10^4^ cells/well) for 48 h. After that, cells were gently washed with cold PBS and fixed by 4% formaldehyde for 10 min. The fixed cells were permeabilized by 0.05% Triton X-100 for 5 min at room temperature. Thereafter, BlockAid™ Blocking Solution was used to minimize the unspecific binding of the antibody. Afterwards, NRF2 primary antibody (Abcam, Cambridge, UK) was diluted (1:50) in blocking buffer and added into the well at 4 °C overnight. On the second day, the primary antibody was washed away with cold PBS for 3 × 10 min and cells were incubated for 1 h at room temperature in the dark with the secondary antibodies (Abcam, Cambridge, UK). A total of 1 μg/mL Hoechst 33342 was used to stain the nuclei of MG-63 cells. Finally, cells were rinsed with cold PBS. Images were acquired using a fluorescence microscope.

### 4.7. Reverse Transcription and Quantitative Real-Time Polymerase Chain Reaction (RT-qPCR)

The total RNA of MG-63 cells was extracted using TRIzol™ (Invitrogen, Waltham, MA, USA) according to the protocol of the total RNA extraction. A total of 2 μg RNA was reversed to cDNA with a First Strand cDNA Synthesis kit (Thermo Fisher, Waltham, MA, USA). RT-qPCR was performed on the Real-Time PCR Detection System (Roche, Basel, Switzerland) using LightCycler^®^ 480 SYBR Green I Master (Roche, Basel, Switzerland) following the manufacturer’s instructions. A standard curve-based method was applied to detect gene expressions, and three housekeeping genes (*HPRT*, *GUSB*, and *PBGD*) were used for expression normalization. Amplification was performed under the following conditions: denaturation at 95 °C for 10 s, renaturation at 55 °C for 30 s, and elongation at 72 °C for 30 s (40 cycles). The fluorescence signal of SYBR Green I was recorded after initial denaturation. After amplification, a melting curve program was executed. The primers used for RT-qPCR are provided in Table 1.

### 4.8. Statistical Analysis

All data were presented as mean ± SD, and GraphPad Prism 9.0 software (GraphPad Inc., San Diego, CA, USA) was used to analyze the data. One-way ANOVA was used for comparison between multiple groups. Tukey’s multiple comparisons test was used for pairwise comparison after ANOVA analysis. *p* < 0.05 was considered statistically significant.

## Figures and Tables

**Figure 1 ijms-24-11975-f001:**
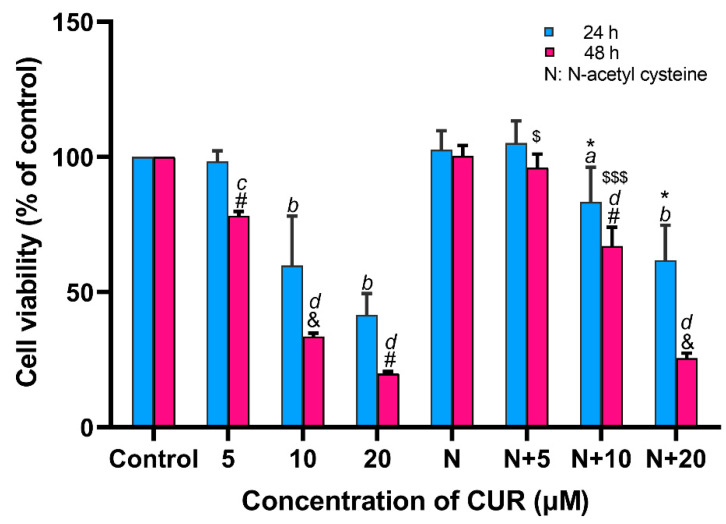
Viability of MG-63 cells treated by curcumin with or without N-acetyl cysteine. *n* = 3 in each group; a: *p* < 0.05 and b: *p* < 0.001 compared with control group in 24 h; c: *p* < 0.05 and d: *p* < 0.001 compared with control group in 48 h; #: *p* < 0.05 and &: *p* < 0.01 comparison between groups in 24 and 48 h with the same CUR concentration; *: *p* < 0.05 comparison between CUR alone groups and NAC with same CUR concentration groups in 24 h; $: *p* < 0.05 and $$$: *p* < 0.001 comparison between CUR alone groups and NAC with same CUR concentration groups in 48 h. CUR: curcumin.

**Figure 2 ijms-24-11975-f002:**
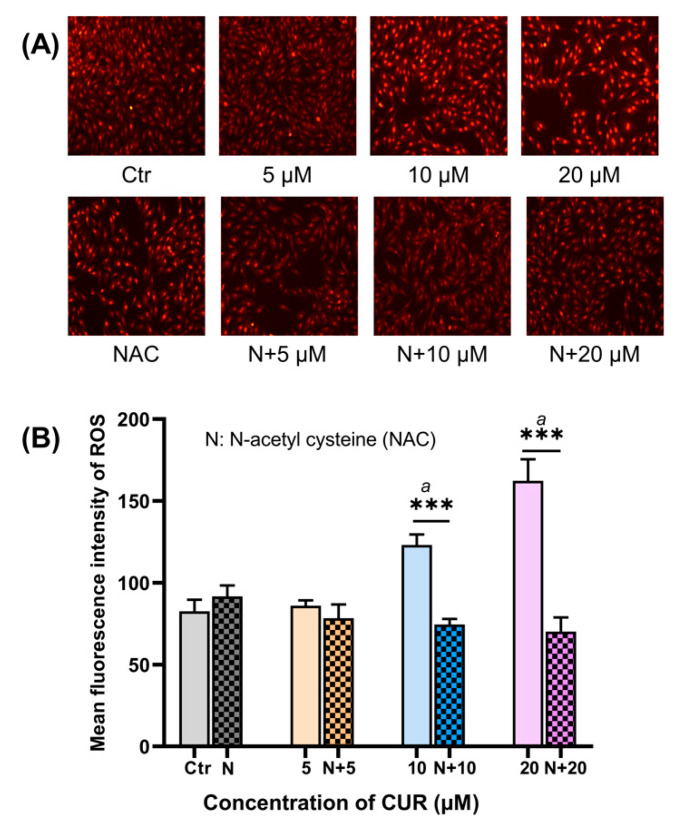
ROS level in MG-63 cells. (**A**) ROS staining in CUR alone or with NAC-treated MG-63 cells; (**B**) mean fluorescence intensity of ROS in each group. *n* = 5 in each group, a: *p* < 0.01 compared with control group, ***: *p* < 0.001.

**Figure 3 ijms-24-11975-f003:**
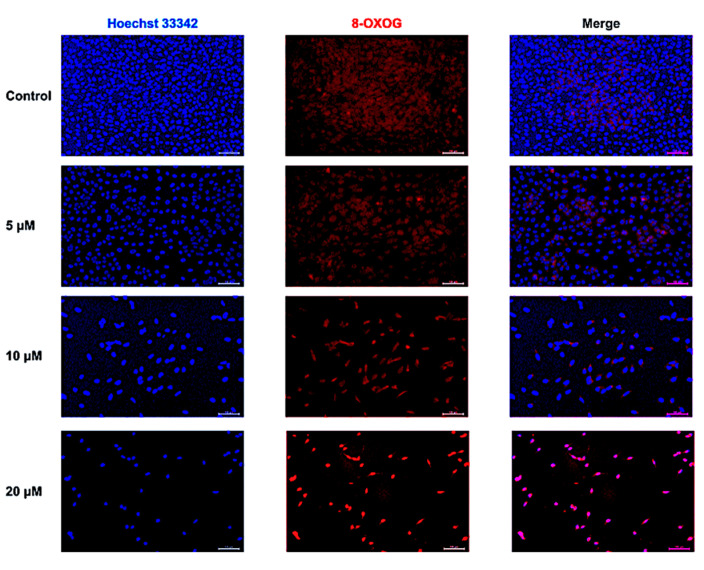
8-OXOG staining of curcumin-treated MG-63 cells. Scale bar = 100 μM.

**Figure 4 ijms-24-11975-f004:**
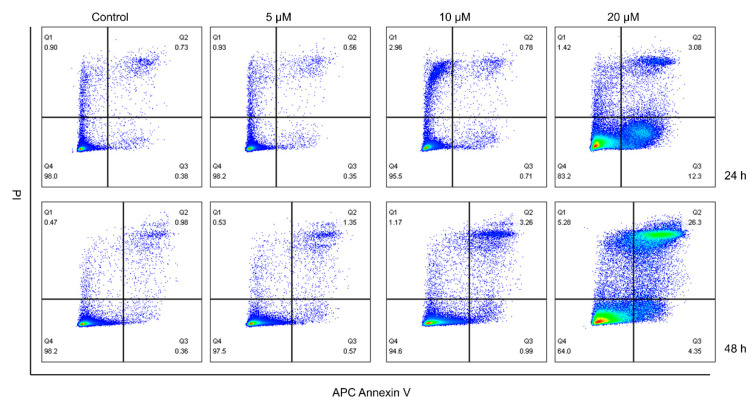
Curcumin induced the apoptosis of MG-63 cells.

**Figure 5 ijms-24-11975-f005:**
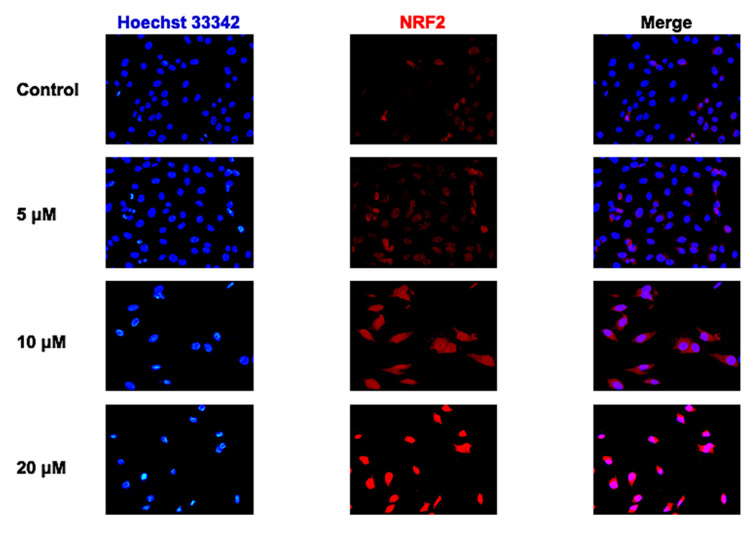
Curcumin stimulated NRF2 nuclear translocation in MG-63 cells. Immunofluorescence staining of NRF2 in MG-63 cells after treatment with curcumin with different concentrations for 48 h (magnification 100×).

**Figure 6 ijms-24-11975-f006:**
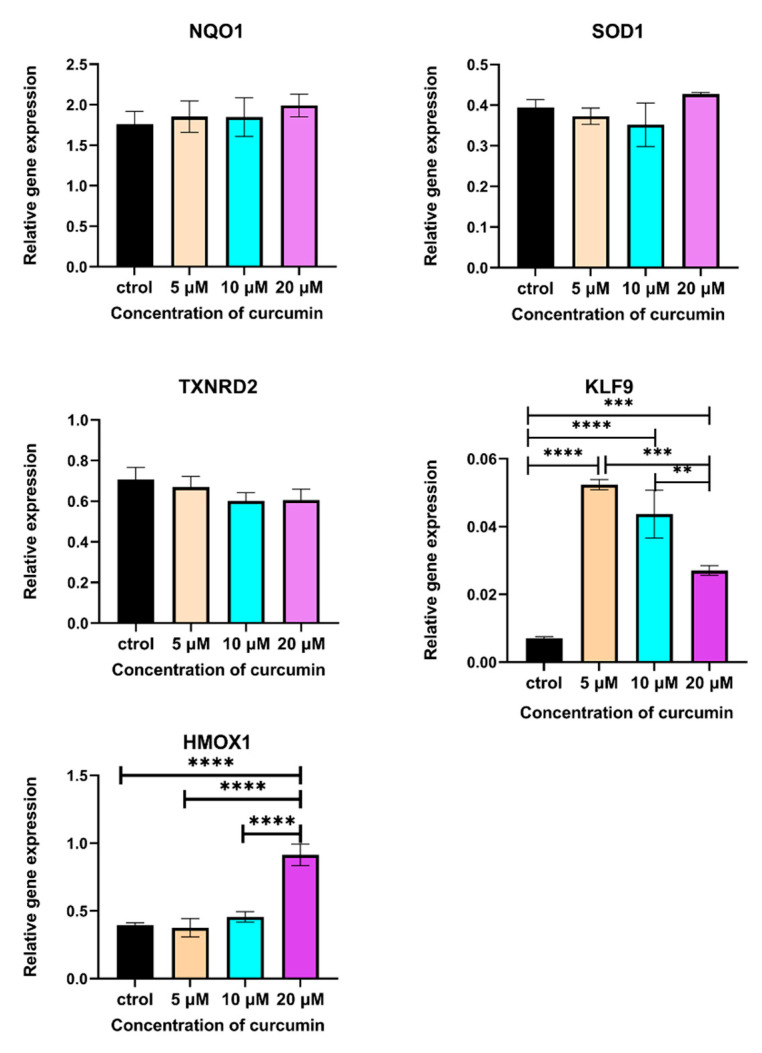
Relative gene expression of NRF2 regulated genes in curcumin-treated MG-63 cells. *n* = 5 in each group, **: *p* < 0.01, ***: *p* < 0.001, ****: *p* < 0.0001.

**Table 1 ijms-24-11975-t001:** List of primers used for gene expression analysis by RT-qPCR.

Target Gene	Primer Nucleotide Sequences	Amplicon Size	Accession No. NCBI Genebank
*HPRT*	FW GCTGACCTGCTGGATTACAT REV CTTGCGACCTTGACCATCT	260	NM_000194
*GUSB*	FW CGCACAAGAGTGGTGCTGAG REV GGAGGTGTCAGTCAGGTA TT	234	NM_000181
*PBGD*	FW TCCAAGCGGAGCCATGTCTG REV CCTGTGGTGGACATAGCAAT	192	NM_000190
*NQO1*	FW CACTGATCGT ACTGGCTCACTC REV ACAGACTCGGCAGGATACTGAA	202	NM_000903
*SOD1*	FW GACTGACTGAAGGCCTGCAT REV TAGACACATCGGCCACACCA	186	NM_000454
*HMOX1*	FW TGCGTTCCTGCTCAACATCC REV CAGCAACTGTCGCCACCAG	233	NM_002133
*TXNRD2*	FW GGAGCATGTTGAGGTCT ATC REV ATCACCTGCGCATAGGAAG	210	NM_006440
*KLF9*	FW CTCCCATCTCAAAGCCCA TT REV AGGTGGTCACTCCTCATGAAG	183	NM_001206

## Data Availability

The data presented in this study are available on request from the corresponding author.

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
