# Peer review of "Paradox: Curcumin, a Natural Antioxidant, Suppresses Osteosarcoma Cells via Excessive Reactive Oxygen Species"

_ijms, 2023, doi:10.3390/ijms241511975_

Round 1

Reviewer 1 Report

Although the manuscript is very interesting, several flaws are present and must be resolved. In particular:

Introduction: since NRF2 plays a key role in this study, the multifaceted role of this transcription factor deserves to be highlithed. In fact, NRF2/KEAP1 signalling plays a key role in several cancerous and non-cancerous diseases as also recently reviewed (PMID: 37296665, 37296999, 36641100, 36289931, 36632321). This is an important point to add since it can further highlight the interesting results obtained by the authors. 

Lines 53-56: it deserves to be mentioned that curcumin showed important effects also in pregnancy complications (see PMID: 33477354)

The number of replicates (N) must be reported in the legend of each figure

Images quality in Figure 3 is very low and must be improved (especially the blu background )

Figure 5: NRF2 stabilization should be investigated by Western Blot. Immunofluorescence is not a quantitative analysis. 

  •  

An accurate revision of syntax and typing errors is recommended

Reviewer 2 Report

  Chu Feng Xu's text describes the role of Curcumin on an osteosarcoma line MG-63. The authors describe the effect of Curcumin on the viability, ROS production and oxidative damage on tumor cells. The authors describe the biological and molecular effects obtained following the administration of Curcumin on tumor cells. The text is interesting, the abstract section summarizes the objectives and results of the study, the introductory part can be reduced. The results section is well illustrated and the discussion reflects the results of the study. The main limitation of the study is represented by the fact that no in vivo experiments are reported and this greatly limits the validity of the study. It would be interesting to study the comparison of the toxic effect of Curcumin together with standard chemotherapy treatments, as reported by previous in vitro and in vivo works. A revision of the English language and the correction of typos in the text is required.                  

It should be improved

Round 2

Reviewer 1 Report

the manuscript has been significantly improved and can be accepted in the present form 

Reviewer 2 Report

The authors fully replied to all criticism

English language is sufficient